

# Automatic procedures for submitting essential climate variables (ECVs) recorded at Italian Atmospheric Observatories to WMO/GAW data centers

Luca Naitza[1], Davide Putero[1], Angela Marinoni[1], Francescopiero Calzolari[1], Fabrizio Roccato[1], Maurizio Busetto[1], Damiano Sferlazzo[2], Eleonora Aruffo[3], Piero Di Carlo[3], Mariantonia Bencardino[4], Francesco D'Amore[4], Francesca Sprovieri[4], Nicola Pirrone[4], Federico Dallo[5], Jacopo Gabrieli[5], Massimiliano Vardè[5,6], Carlo Barbante[5], Paolo Bonasoni[1], and Paolo Cristofanelli[1]

[1]CNR–ISAC, National Research Council – Institute of Atmospheric Sciences and Climate, Bologna, Italy
[2]ENEA, SSPT-PROTER-OAC, Lampedusa, Italy
[3]Chieti University, Chieti, Italy
[4]CNR–IAA, National Research Council – Institute of Atmospheric Pollution, Rende, Italy
[5]CNR–IDPA, National Research Council – Institute of Dynamics for Environmental Processes, Mestre-Venezia, Italy
[6]Dept. of Chemical and Pharmaceutical Sciences, Ferrara University, Ferrara, Italy

*Correspondence to:* P. Cristofanelli (p.cristofanelli@isac.cnr.it)

**Abstract.** In the framework of the National Project of Interest NextData, we developed procedures for the automatic flagging and formatting of trace gases, atmospheric aerosol and meteorological data to be submitted to the Global Atmosphere Watch program of the World Meteorological Organization (WMO/GAW). In this work, we describe a first prototype of a centralized system to support Italian atmospheric observatories towards a more efficient and objective data production and subsequent

5  submission to WMO/GAW World Data Centers (WDCs). In particular, the atmospheric variables covered in this work were observations of near-surface trace gas concentrations, aerosol properties and (ancillary) meteorological variables, which are under the umbrella of the World Data Center for Greenhouse Gases (WDCGG, see https://ds.data.jma.go.jp/gmd/wdcgg/), the World Data Center for Reactive Gases, and the World Data Center for Aerosol (WDCRG and WDCA, see http://ebas.nilu.no). For different Essential Climate Variables (ECVs), we developed specific routines for data filtering, flagging, format harmonization,

10  and creation of data products (i.e., plots of raw and valid-corrected-averaged ECV data and internal instrument parameters), useful for detecting instrumental problems or particular atmospheric events. A special suite of products based on the temporal aggregation of valid ECV data (like the "calendar" or "timevariation" products) were implemented for quick data dissemination towards stakeholders or citizens. Currently, the automatic processing of data is active for a subset of ECVs at 4 measurement sites in Italy. The NextData system does not generate "consolidated" data to be directly submitted to WDCs, but it represents

15  a valuable tool to facilitate data originators towards a more efficient data production for those data streams. Our effort is expected to accelerate the process of data submission to WMO/GAW or to other reference data centers or repositories, as well as to make the data flagging more "objective", which means that it is based on a set of well-defined selection criteria and not strictly related to the subjective judgment of station operators. Moreover, the adoption of automatic procedures for data



flagging and data correction allows to keep track of the process that led to the final validated data, and makes data evaluation and revisions more efficient.

# 1  Introduction

The monitoring of trace atmospheric constituents in the lower troposphere still represents a fundamental activity to assess their

long- and short-term variability, to investigate the underlying processes and to assess the impact of natural and anthropogenic sources into the atmosphere. The Global Atmosphere Watch program of the World Meteorological Organization (WMO/GAW) coordinates a global network of surface stations to understand and control the increasing influence of human activity on the global atmosphere (WMO, 2017). The backbone of the WMO/GAW is a global network of more than 400 surface stations, performing routine observations of atmospheric constituents in the troposphere/stratosphere. Atmospheric stations belonging

to the WMO/GAW network are requested to adopt standard operating procedures (SOP), to perform quality assurance/quality check (QA/QC) actions and routinely submit data to specific World Data Centers (WDCs) covering 6 focal areas (i.e., atmospheric aerosols, greenhouse gases, selected reactive gases, ozone, UV radiation, precipitation, and chemistry). In Italy, the NextData project was aimed at creating a network in mountain and remote areas, based on atmospheric observatories for the monitoring of atmospheric composition and ancillary data (meteorological parameters and solar radiation). The main goal of

this network is to investigate the processes which influence the variability of air pollutants and climate-altering compounds to contribute towards a better assessment of the related impacts to mountain ecosystems and climate in the Mediterranean basin. The network comprises five high-mountain atmospheric observatories (Fig. 1): Monte Cimone (CMN, northern Apennines, the only WMO/GAW global station in Italy; 2165 m a.s.l.), Plateau Rosa (PRS, western Alps, WMO/GAW regional station; 3480 m a.s.l.), Col Margherita (MRG, eastern Alps; 2550 m a.s.l.), Monte Portella–Campo Imperatore (CMP, central Apennines;

2401 m a.s.l.), and Monte Curcio (CUR, southern Apennines, WMO/GAW regional station; 1796 m a.s.l.). In addition to these observatories, the WMO/GAW regional stations Capo Granitola (CGR, south-western Sicily) and Lampedusa (LMP, central Mediterranean Sea), provide complementary information on the background conditions of the Mediterranean basin marine boundary layer (Fig. 1).

Six of these observatories are already part of international projects/research programs for the monitoring of Essential Cli-

mate Variables (ECVs). More specifically, in the framework of the WMO/GAW activities, observations of greenhouse and reactive gases are carried out at PRS, CMN, CUR, CGR and LMP (Table 1). Moreover, PRS, CMN and LMP also started the labeling processes to be included in the European Research Infrastructure ICOS (Integrated Carbon Observation System, http://www.icos-ri.eu, see Hazan et al., 2016). Measurements of physical properties of atmospheric aerosols are performed at CMN, CUR, and CGR (Table 1). CMN and CGR are part of ACTRIS-2 (Aerosols, Clouds and Trace gases Research In-

frastructure, https://www.actris.eu/, see FMI, 2017), which aims at consolidating high-quality observations of aerosols, clouds and trace gases in Europe. MRG and CUR were part of GMOS (Global Mercury Observation System), a global observational network providing comparable data on mercury levels in ambient air and deposition (see Cinnirella et al., 2014; Sprovieri et al., 2016). Within GMOS, a web-based system for QA/QC has been developed in order to check raw data related to atmospheric





mercury (D'Amore et al., 2015). Over 2017–2020, both MRG and CUR will be involved in the iGOSP (Integrated Global Observing Systems for Persistent Pollutants, http://www.igosp.eu), strand 3 of ERA-PLANET project (Tsinganos et al., 2017), funded in the framework of the Horizon2020.

To contribute to the implementation of this network, one action carried out by NextData was to set-up a system for the auto-
matic processing of the ECVs data observed at these measurement sites, thus supporting the participation of these atmospheric observatories in the WMO/GAW activities. In particular, the ECVs covered in this work primarily focus on near-surface trace gases, aerosol properties and (ancillary) meteorological parameters, which are under the umbrella of the World Data Center for Greenhouse Gases (WDCGG, see https://ds.data.jma.go.jp/gmd/wdcgg/), the World Data Center for Reactive Gases, and the World Data Center for Aerosol (WDCRG and WDCA, see http://ebas.nilu.no). The goal of this activity is to support the
measurement sites towards an optimization of the data creation process, which is a pre-requisite for a fast and efficient data publication. Indeed, due to the large amount of data recorded at the measurement sites, it is not efficient to perform the data validation/flagging by the so-called "visual inspection" and by manual manipulation of data files. Moreover, there is a widely recognized need for the provision of ECVs in near-real time or real-time mode for a number of applications (data assimi-lation, atmospheric model verification, early warning systems, see, e.g., Wagner et al., 2015), which imply the delivery of
quality-assessed data with well-defined data formats.

For these reasons, we hereby developed automatic procedures for ECVs data flagging, averaging, and formatting. The im-plementation of such automatic procedures will represent a powerful resource to help the researchers in spending more time for the scientific purposes, rather than data verification and formatting. Besides making the data creation process faster and favoring a timely data submission, the adoption of standardized validation procedures will also assure a more uniform flagging
of data, as well as the possibility to trace back the actions which led to data validation (i.e., data revisions will be easier). It must be clearly stated that these actions would not overcome the QA/QC activity carried out at the topical/thematic centers of integrated initiatives (e.g., ICOS, ACTRIS), nor at WMO/GAW WDCs.

Currently, the automatic processing of data is active for a subset of ECVs and measurement sites (see Table 2). In particular, the CMN observatory (Cristofanelli et al., 2018) was selected as a "proof-of-concept site", due to the large number of ECVs
observed and the large variety of data formats produced by the measurement systems. On a daily basis, raw data from measure-ment sites are transferred to a server located at CNR–ISAC headquarters (HQs) in Bologna, Italy, for automatic data processing and storage. The automatic data processing encompasses a preliminary harmonization of file formats, which is the pre-requisite for the subsequent data flagging, data aggregation (to common temporal frames: 1 and 60 minutes), and final harmonization of files according to WMO/GAW WDCs formats. The automatic data processing also encompasses the creation of many data
reports (updated daily), which provide an overview of the instruments and data behavior, to support both the quality control of data, as well as the data inspection for scientific or operational purposes (i.e., the identification of events of interest, or to perform preliminary data analysis). Specific procedures have been developed for each ECV and for specific instruments (i.e., one processing chain for each instrument type) used at the considered stations: Table 2 summarizes the list of these specific instruments, together with the formats used for file creation. All the procedures have been implemented in "R" Language and
Environment for Statistical Computing (R Development Core Team, 2008). It is important to note that the NextData routines,





even if implemented for a specific suite of instruments, can be virtually adapted to other instruments using the same principle of operation/detection.

In the following, through explanatory examples, we will provide a detailed description of the architecture and workflows of the automatic routines. Due to the recent change of file format requested by WMO/GAW WDCGG, the processing chain for

$CO_2$, $CH_4$ and CO will be implemented as soon as possible (at the moment, flagging and averaging procedures are available, but they are not yet implemented in the automatic processing chain), while a trial system for reactive gases and aerosol properties (under WMO/GAW WDCRG and WDCA) is already on-line.

## 2   Description of the automatic data processing

To increase the inter-operability of the automatic procedures, the sequential steps of the workflow are the same for all the

different ECVs, and specific modules or functions have been developed to be inserted in the programming code to increase flexibility of usage:

1. data file collection from observatories;

2. data formatting of raw files coming from stations;

3. data check and flagging;

4. data correction (if needed);

5. data aggregation (time averaging) and flagging;

6. data formatting;

7. creation of data reports.

By this 7-steps process, three different data levels are produced according to WMO/GAW WDCRG and WDCA data reporting

guidelines (see also https://ebas-submit.nilu.no/Submit-Data/Data-Reporting):

- Level-0: annotated raw data; contains all parameters and variables provided by the instrument; contains all parameters/info needed for processing to final value; "native" time resolution;

- Level-1: data processed to final variable (calibration and correction implemented to data series), invalid data and calibration episodes removed, "native" time resolution, correction to standard temperature and pressure (i.e., 273.15 K, 1013.25

hPa) if necessary;

- Level-2: data aggregated to hourly averages, atmospheric variability quantified by standard deviation or percentiles.

In the following, we will describe each step of the processing chain. To provide an explanatory example, we will describe the processing chain and the routines developed for the Thermo 42i-TL, a state-of-art instrument for the continuous determination





of nitrogen monoxide (NO) and nitrogen dioxide ($NO_2$) mixing ratios, based on chemiluminescence detection (CLD) and equipped with a blue light converter (BLC). This instrument belongs to a class of instruments (Thermo "i-series") widely diffused among the NextData measurement stations for trace gas measurements and, due to the complexity of steps necessary to obtain the final data, it represents an effective case study for implementing mature QA/QC routines. Fig. 2 reports the different steps of the data processing for this analyzer.

## 2.1 Automatic processing of ECVs: data collection and formatting of raw files

The instrumental raw data are transferred (at least) once a day from the measurement stations to a server located at CNR–ISAC HQs in Bologna. The files stored in the CNR–ISAC server already contain information of the measured quantities in geophysical units, as well as internal diagnostic parameters used for automatic QA/QC. During the development phase, different transfer strategies are applied, as a function of the specific requirements of the measurement stations. For instance, CMN data files are downloaded from the station server, while MRG data files are uploaded by the station server to the CNR–ISAC server. To facilitate the participation of the stations we let each station decide which strategy to adopt for data transfer during this development phase.

All the raw data files are centralized to the CNR–ISAC server as a function of their origin station, ECV class (trace gases, aerosol properties, meteorology) and specific ECV (see Table 3). Then, they are processed to obtain a homogeneous file system in terms of nomenclature and format, which is read by the processing chain. For the file name, we adopted the following coding:

$$SSS\_PPP\_YYYYMMDD\_TTT.dat \tag{1}$$

where SSS is the station code, PPP is the ECV code (see Table 3 for the complete list of adopted codes), YYYYMMDD is the file production date, and TTT is the native time resolution of the measurements (i.e., 1-min: "01M", 30-min: "30M", 60-min: "60M"). These homogenized data files refer to UTC (Coordinated Universal Time), and they have a common structure for what concerns date and time: year (YYYY), month (MM), day (DD), hour (HH), minute (MIN) and the decimal date for each measurement (DEC_DATE), considered as fraction of time from the reference point (i.e., January 1st of each year, at 00:00 UTC).

## 2.2 Automatic processing of ECVs: data check and flagging

The data inspection consists of a series of checks that are automatically performed on data with native time resolution (usually 1-min), with the aim of referring each data record to a set of codified flags (which indicate whether a measurement is valid or not). Table 4 reports the list of flags adopted for $NO_x$ measurements, as defined in the framework of ACTRIS-2 project (see https://ebas-submit.nilu.no/Submit-Data/Data-Reporting/Templates/Category/Trace-Gases/NOx).

The first step of the automatic data check consists in the identification of the different measurement modes: "sample" (when ambient air is measured), "calibration" (when air from one or more laboratory standards is measured), "zero" (when a gas mixture scrubbed by the molecules to be quantified is measured, typically to determine the instrumental zero off-set for routine



quality checks) and "span" (when dry air enriched by a specific amount of the molecules to be quantified is measured, typically to point out changes in the instrumental sensitivity). Systematic variations with time of the zero offset and the span factor are used to timely detect instrumental problems, while a "full" calibration is used to link the measurement to a reference calibration scale hosted by a central laboratory and to verify measurement linearity. Depending on the ECV and the instrument, the zero

source can be either tanks with a dry gas mixture or a generator producing clean dry air, while span sources can be a tank with assigned mixing ratios for a specific chemical, a permeation tube with a precise and stable emitting rate for a specific chemical, or an internal generator (like a UV-lamp in case of $O_3$). The identification of the measurement mode, which leads to the attribution of a specific flag to the concurrent data record, is fundamental for two reasons: (i) data affected by calibrations or quality checks must be discarded during time averaging processes, and (ii) data recorded during calibrations or quality

checks are used to obtain correction factors or quality control metrics. The measurement periods affected by "calibrations" or "zero/span" checks are selected considering two general cases: (i) by analyzing the variability of internal diagnostic parameters of analyzers or external "calibration units" (e.g., some instruments provide the information related to their current "mode" to the acquisition system), or (ii) by searching the existence of a log-file that indicates the start and end times of QC exercises.

The second step is the analysis of the variability of instrumental data (both related to internal diagnostic parameters and

to the measured ECV). Such step is based on general criteria, but, at the same time, it is adapted as a function of specific measurement stations and ECVs. The following checks are implemented in the data control process:

– Diagnostic/instrumental checks: the internal diagnostic parameters (e.g., temperatures, flows, pressures) are compared with their typical ranges, which are reported on the instrument manuals. For each measurement, if at least one parameter fails a check, the data is flagged as invalid (see, e.g., Table 5 for NO and $NO_2$ threshold values at CMN);

– Plausibility checks: the measurement periods with values exceeding the expected variability are selected and identified as "outliers". The allowed variability ranges are defined as a function of the measurement stations (e.g., the plausible range for atmospheric pressure at a surface station like CGR is different from that of a mountain station like MRG). Currently, two different processes for the definition of these ranges are considered. The first one is the adoption of fixed threshold values, defined upon existing literature, and in collaboration with scientists in charge of the instruments. The second

process is related to the on-line calculation of variable threshold values based on statistical data analysis over specific time periods (1 hour, 1 day, 1 month, or a full year), e.g., percentiles of the data-set or confidence intervals like $n$-times the standard deviation above or below the average values. In this context, a specific calculation has been implemented to detect outliers, and applied to raw data with a valid numflag (no calibration, no sampling failures/interruptions) for the aerosol variables (i.e., NEPH, MAAP, OPC, CPC). Following an approach similar to Huang et al. (2016), outliers are

identified (and flagged) if:

$$|x_i - RM_i| \geq nRSD_i \tag{2}$$

where $x_i$ is the $i$th measurement, $RM_i$ and $RSD_i$ are the running mean and running standard deviation for the $i$th record on a user-defined window (e.g., 3 days), and $n$ is a user-defined value to be set according to the instrument and type of the





station. For CO, $CH_4$, and $CO_2$ data, the outlier detection procedure based on the standard deviation of the background (SD, see El Yazidi et al., 2018) was implemented;

- Variability checks: verification of the variability (i.e., rate of change with respect to time) of the observed ECV. Depending on the considered ECV and the characteristics of each site, a range for maximum ECV variability is defined
(typically on hourly basis, see, e.g., Table 5 for NO and $NO_2$ threshold values at CMN).

## 2.3 Automatic processing of ECVs: data calibration and correction (Level-1 production)

As specified in Sect. 2, Level-1 refers to the data set containing only "valid" records with calibration and corrections applied (i.e., data with "not-valid" or "calibration" numflags are removed).

For $NO_x$, the major difficulty in this task is related to the use of the information provided by automatic calibration in the
processing chain (zero and span). CLD instruments directly detect and quantify only NO; therefore, it is necessary to convert $NO_2$ to NO to quantify $NO_x$ (and finally $NO_2$). The commercially available instruments for air-quality monitoring are usually equipped with a Molybdenum (Mo) heated converter. However, this set-up is not recommended by GAW (2011) in rural/remote locations, since this kind of detector is not selective to $NO_2$: it also converts other oxidized nitrogen compounds, such as nitric acid ($HNO_3$), peroxyacetyl nitrate (PAN), and other organic nitrates (Steinbacher et al., 2007). For these reasons, the CMN
instrument is equipped with a photolytic converter (Blue Light Converter, Teledyne, USA), which uses an UV light source to selectively convert $NO_2$ to NO. Since for some instruments BLC conversion efficiency (Sc) can be significantly lower than 100%, it is paramount to derive the actual value of Sc. The Sc obtained by a gas phase titration carried out during the calibration is then used to correct the $NO_2$ reading and to obtain the actual $NO_2$. The calibration process is composed of the following steps:

1. Sampling of zero air: the NO reading is used to calculate instrumental zero-offset (bkg_NO, bkg_$NO_x$);

2. Sampling of span air (obtained by diluting 5 ppm of NO in $N_2$ to about 100 ppb): the NO and $NO_x$ readings are used to calculate the NO and $NO_x$ span factors (coeff_NO, coeff_$NO_x$) for obtaining corrected NO (NO_elab1) and $NO_x$ ($NO_x$_elab1);

3. After the determination of the new calibration factors for NO (zero offset and span factor), $O_3$ is added to the mixture,
so that about 80% of the NO is titrated (gas phase titration–GPT);

4. After stabilization, the data for NO (NO_elab1_gpt) and $NO_x$ ($NO_x$_elab1_gpt) are recorded and used to calculate converter efficiency (Sc, see below);

5. Sampling of zero air to purge the instrument after calibration.

Then, Sc is calculated as follows. The effective $NO_2$ amount produced by GPT results from:

$$NO_2 = NO\_elab1\_span - NO\_elab1\_gpt \tag{3}$$




The NO$_2$ amount converted by BLC is calculated by:

$$(\mathrm{NO_x\_elab1\_gpt} - \mathrm{NO\_elab1\_gpt}) - (\mathrm{NO_x\_elab1\_span} - \mathrm{NO\_elab1\_span}) \qquad (4)$$

Accordingly, Sc is calculated by:

$$\mathrm{Sc} = \frac{(\mathrm{NO_x\_elab1\_gpt} - \mathrm{NO\_elab1\_gpt}) - (\mathrm{NO_x\_elab1\_span} - \mathrm{NO\_elab1\_span})}{\mathrm{NO\_elab1\_span} - \mathrm{NO\_elab1\_gpt}} \qquad (5)$$

For each step of the calibration event, only the last 10 minutes of data are considered to allow system stabilization. The values of NO, NO$_2$, and NO$_x$ during the different steps of calibration, together with calculated zero offsets, span coefficients, and Sc values are stored in an internal table and plotted to allow verification of instrumental performance. For the first four months of 2018, Fig. 3 reports the different calibration coefficients and NO-NO$_2$-NO$_x$ values measured at CMN during the calibration steps. The "calibrated" NO$_2$ is obtained by subtracting the "calibrated" NO from the "calibrated" NO$_x$. Several threshold values are applied to the calculated calibration factors (Table 5): if the check of calibration factors ("Zero offset", "Span coeff", "Sc") against defined threshold values fails, the last calibration factors successfully calculated are retained for data correction.

Concerning the data correction, at the current stage only the correction of the NO night-time bias is implemented. For each date we calculate the night-time (00:00–04:00) average value. Under excess of O$_3$ (like in a remote or rural region during night-time), NO must be completely titrated by the reaction:

$$NO + O_3 \rightarrow NO_2 + O_3 \qquad (6)$$

In these conditions, NO is expected to decrease below the instrumental detection limit. If this does not happen, a "night-time" zero offset correction is calculated by smoothing the "night-time" NO reading over 2 days, and by subtracting the "night-time" offset from the Level-1 data series.

## 2.4 Automatic processing of ECVs: data aggregation (Level-2 production)

Basing on the data screening and corrections applied in the previous steps (i.e., coding of Level-0 and Level-1 data), 1-min data are aggregated to hourly (60-min) average values for obtaining Level-2 data. For time aggregation, only data with a valid numflag are considered. This means that also data "under detection limit" are used for data aggregation. Besides average values, as a function of the different ECV, other statistical parameters are computed, as required by WDC data formats. In case less than 50% of data are used for the calculation of the hourly mean value, the hourly data is properly flagged (i.e., 0.390). Moreover, a set of three functions for creating, checking and aggregating numflags was developed. In the following, we will provide a brief description for each of them:

- "nf_aggreg": this function adds the desired numflag value to the already existing ones. E.g., it adds 0.456 to the already existing 0.640, so that is becomes 0.456640;

- "nf_val_check": this function easily checks if the numflag given as input is valid or not. The validity is checked by using a valid/not valid list given as input. In the current version of our system, the original numflag list provided by EBAS



is implemented (https://ebas-submit.nilu.no/Submit-Data/List-of-Data-flags). This function works efficiently also with composed numflags (e.g., 0.640980). In this case, if at least one of the "internal" numflags is invalid, the composed numflag will be invalid;

- "nf_lev2": this function builds the hourly numflag for the Level-0 or Level-1 data given as input, together with the
start_time for both Level-1 (or Level-0) and Level-2 data. As also specified in the WDCRG and WDCA guidelines (see http://ebas-submit.nilu.no), all of the informative (valid) numflags that occurred at any time during the average period need to be copied (and aggregated, if necessary) to form the hourly numflag value (e.g., if two minutes have the 0.640 numflag, and five minutes have the 0.410 numflag, the final built numflag would be 0.410640).

Figure 4 shows an overview of $NO_2$ mole fraction measured at CMN on January 2018 (this is an extract of the opera-
tional product CMN_NO2_2018_01_MONTHLY_GRAPH_20180730.png, see Sect. 2.6.3). Here we reported the time series of Level-2 data (i.e., 1-hour calibrated and corrected average values), the concurrent data flags and a comparison between "final" Level-2 data and raw data from the instrument. When looking at the comparison of "native" instrumental output (1-min resolution) and Level-2 data, one can see which data were discarded before averaging, leading to the appearance of missing values in the Level-2 dataset (i.e., numflag equal to 0.999).

**2.5    Automatic processing of ECVs: data formatting**

To optimize the interoperability of the data system, the processed data are formatted in agreement with the guidelines of the WMO/GAW data-centers. The greenhouse gases ($CO_2$, $CH_4$) and the carbon monoxide (CO) data are created in agreement with formats and metadata indicated by the WDCGG, as reported in "Revision of the WDCGG Data Submission and Dissemination Guide" (GAW Report No. 188). "Near-surface" reactive gases, aerosol properties and meteorological parameters are formatted
following the NASA-AMES standard, as indicated by WMO/GAW WDCRG and WDCA. This format is based on the textual format ASCII NASA-AMES 1001 with additional metadata (as a function of the different ECVs). The templates to be used for the different ECV can be found at the web page http://ebas-submit.nilu.no/Submit-Data/Reporting-Templates/all-templates-temporary. The files created by the automatic processing chain contain all of the observations carried out during a full solar year; they are updated on a daily basis.

**2.6    Automatic processing of ECVs: data products**

To support measurement stations in carrying out the QA/QC checks, a suite of products (i.e., data plots) is produced by the automatic data processing chain. The data products are updated on a daily basis, by using specific routines. To this aim, some specific functions of the "OpenAir" package (Carslaw and Ropkins, 2012) are also used. The data products are arranged as a function of different time windows: daily, monthly, seasonal and yearly. In total, 9 data reports are operationally produced for
each ECV. Data reports are ".png" files identified by the following name code:

$$SSS\_PPP\_yyyy\_mm\_PERIOD\_TYPE\_YYYYMMDD.png \tag{7}$$





where SSS is the station code, PPP is the ECV code (see Table 3), yyyy_mm identifies the time validity of the product (for data reports related to a full calendar year the code yyyy_01 is conventionally adopted), PERIOD is the period of time spanned by the data reports (i.e., "DAILY", "MONTHLY", "SEASONAL", "SEMESTER", "ANNUAL"), TYPE denotes the class of data product (i.e., "GRAPH", "TIMEVARIATION", "CALENDAR", see Table 6), and YYYYMMDD is the file production date.

For example, the code CMN_NO2_2018_01_MONTHLY_GRAPH_20180730.png identifies the monthly data product n. 1 (see below) created on July 30th, 2018, for $NO_2$ measured at CMN on January 2018 (CMN_NO2_2018_01). In the following, we will provide a brief overview of the data reports, while in the Supplementary Material we provide specific examples for each product.

### 2.6.1   Daily data products (SSS_ECV_yyyy_mm_DAILY_GRAPH_YYYYMMDD.png)

Description: These products are based on automated plots generated daily, by using native time resolution of data. They provide a time series (along a full calendar day) of ECV raw and corrected data, together with instrumental diagnostic parameters and Level-0 flags.

Aim: The main purpose is to have a daily diagnostic about measurement status, as well as a high-resolution quick-view of the ECV variability on a daily time frame (i.e., last 24 hours).

### 2.6.2   Monthly data products 1 (SSS_ECV_yyyy_mm_MONTHLY_GRAPH_YYYYMMDD.png)

Description: These products are based on automated plots generated daily, and comprise corrected/calibrated averaged ECV data (Level-2), informative numflags (Level-0), as well as raw data and internal instrumental parameters (Level-0) over a full calendar month. For each month, a reference line describing the ECV average value, as well as the indication of the lowest and the highest ECV values, are provided . This product also provides, embedded in the plot plate, a table with basic statistical
parameters for the ECV Level-2 data.

Aim: The main purpose is to provide diagnostic about instrument performance (e.g., detecting medium-term instrumental drifts able to affect measurements) on a monthly time frame, as well as to give an overview of Level-2 data and their comparison with Level-0 data.

### 2.6.3   Monthly data products 2 (SSS_ECV_yyyy_mm_MONTHLY_TIMEVARIATION_YYYYMMDD.png)

Description: These products use the "timeVariation" function of the "OpenAir" package (Carslaw and Ropkins, 2012) on valid Level-2 data. For each month, plots representing the average diurnal variation of the considered ECV (with 95% confidence intervals) for the entire month and for each day of the week are provided.

Aim: This product gives information about typical diurnal variability of the selected ECV. The main purpose is the operational data reporting and QA/QC (identification of anomalous behavior for the selected ECV potentially due to experimental
problems).





### 2.6.4 Monthly data products 3 (SSS_ECV_yyyy_mm_SEMESTERn_GRAPH_YYYYMMDD.png, n=1,2)

Description: These products are based on automated plots generated daily using corrected and averaged ECV Level-2 data. They provide the time series of corrected ECV values for each month over a semester (i.e., January–June, and July–December). For each month, a reference line describing the ECV average value, as well as the indication of the lowest and the highest ECV

values, are provided. This product also contains, embedded in the plot plate, a table with basic statistical parameters for the ECV Level-2 data.

Aim: The main purpose is the operational data reporting by providing multi-months ECV time series, also useful to detect the occurrence of "special" events.

### 2.6.5 Seasonal data products (SSS_ECV_yyyy_mm_SEASONAL_TIMEVARIATION_YYYYMMDD.png)

Description: These products are based on automated plots generated daily using corrected and averaged ECV data (Level-2). For each season (DJF, MAM, JJA, SON), they provide the average diurnal variation of corrected ECV values and average weekly variation. For each analysis, the statistical 95% confidence interval of average values is provided.

Aim: These products provide information about typical diurnal variability of the selected ECV for each season: the main purpose is the operational data reporting and QA/QC (i.e., identification of anomalous diurnal behaviors potentially due to

experimental problems).

### 2.6.6 Quarterly data products (SSS_ECV_yyyy_mm_SEASONAL_GRAPH_YYYYMMDD.png)

Description: These products are based on automated plots generated daily using corrected and averaged ECV (Level-2) data. For each quarterly period (JFM, AMJ, JAS, OND), the time series of ECV, together with the histogram and the power density function of the ECV data, are reported.

Aim: These products give information about statistical variability of the selected ECV for each quarterly period: the main purpose is the operational data reporting and QA/QC (i.e., identification of data population distribution changes or outliers potentially related to experimental problems).

### 2.6.7 Yearly data products 1 (SSS_ECV_yyyy_mm_ANNUAL_GRAPH_YYYYMMDD.png)

Description: This product is based on automated time series generated daily by using corrected and averaged ECV data (Level-

2) for the whole year. Additionally, flags (Level-1) and internal diagnostic parameters (Level-0) are provided. Statistical information about ECV variability and the percentage of valid data is also given.

Aim: The main purpose is to have a medium-term diagnostic about corrected ECV data, useful also to detect the occurrence of "special" events. Moreover, internal instrumental parameters are plotted for QA/QC purposes. This would allow the station Principal Investigator (PI) to detect possible instrumental changes or drifts occurring over several months.





### 2.6.8   Yearly data products 2 (SSS_ECV_yyyy_mm_ANNUAL_CALENDAR_YYYYMMDD.png)

Description: This product uses the "calendarPlot" function of the "OpenAir" package (Carslaw and Ropkins, 2012) on Level-2
corrected data.

Aim: It provides the average daily values throughout the year for the selected ECV. The main purpose is operational data

reporting and quick-view about ECV data availability/coverage at the station.

### 2.6.9   Yearly data products 3 (SSS_ECV_yyyy_mm_ANNUAL_TIMEVARIATION_YYYYMMDD.png)

Description: These products are based on automated plots generated daily using corrected and averaged ECV (Level-2) data.
They provide the average diurnal variation of corrected ECV values, their annual cycle and average weekly variation calculated
over a whole year. For each analysis, the statistical 95% confidence intervals of average mean values are provided.

Aim: These products give information about the typical variability (over different time scales: diurnal, weekly, and annual)
of the selected ECV for the whole year. The main purpose is the operational data reporting and QA/QC (i.e., identification of
anomalous diurnal behaviors potentially due to experimental problems).

### 3   Summary and perspectives

To favor the integration of a background network in Italy, we implemented a first prototype of a centralized system to support

measurement stations in data production and subsequent submission to reference data centers (WMO/GAW, or other research
programs). Indeed, there is an increasing need for ECV data with well-assessed data quality (WMO, 2016), but the increasing
data quality demand and the request of specific data format often faces against limited human resources at the atmospheric
observatories. Leading research infrastructures (e.g., ICOS) or pan-European projects (e.g., ACTRIS-2) already implemented
centralized facilities to process raw data, to perform quality controls and to provide standardized data outputs, but these efforts

are limited to a set of selected participating stations and to a list of "core" ECVs. It should be clearly stated that our system
would not overcome station manager or instrument operators in their responsibility for the final validation and data submission
to data centers. The NextData system does not generate "consolidated" data to be directly submitted to data centers; however,
it represents a valuable tool to facilitate data originators towards a more efficient data production for those data streams which
are not covered by already-existing services.

25       By using the "R" Language and Environment for Statistical Computing (R Development Core Team, 2008) we developed,
for each of the ECVs listed in Table 2, specific routines for data filtering, flagging, formatting, and creation of data products
useful for detecting instrumental problems or particular atmospheric events. A special suite of products (like the "calendar" or
"timevariation" products, created by a specific function of the "OpenAir" package) have been thought to be useful for a quick
data dissemination towards stakeholders or citizens. Our effort is expected to improve data quality, to accelerate the process of

data submission to WMO/GAW WDCs (in this perspective, we will encourage the adoption of these procedures also by other




measurement stations, not directly related to NextData) and to make the data flagging more "objective", which means that it is based on a set of well-defined selection criteria and not strictly related to the subjective judgment of station operators.

Further improvements must be implemented in the current prototypal version of the system. In this current implementation phase, specific routines have been completed for the different stations and ECV classes, to harmonize the different raw files to
a common format that can be used by the processing chain. In the future, it is foreseen that data, even from new stations, can be provided to CNR–ISAC server already in the homogenized format. Further data correction procedures must be realized for some specific ECVs (e.g., water vapor and $O_3$ interferences to NO and $NO_2$, see Gilge et al., 2014)). More work should be done concerning the automatic flagging of outliers that, for some species (e.g., $O_3$, NO, $NO_2$, $SO_2$), still relies on fixed thresholds. However, a not negligible amount of work has been carried out also in the direction of setting-up an automatic function for
outlier selection, which is already implemented in the processing chain of aerosol parameters, and in the preliminary procedures for CO, $CH_4$, and $CO_2$.

As detailed in Sect. 4, the developed routines will be freely accessible on the NextData website: the interaction and contribution of the scientific community will certainly represent a possibility for improving them.

## 4   Routine and data availability

All the routines implemented for the different steps involved in the automatic processing of data are written by using the "R" Language. Once finally validated, these routines will be freely accessible on the NextData website. A subset of historical data about ECVs reported in this work are freely available on the platform MOVIDA-Multistat (http://shiny.bo.isac.cnr.it:3838/plot-multistats-en/), implemented in the framework of NextData. Metadata and data obtained thanks to the direct support of Next-Data can be found at: http://geonetwork.igg.cnr.it

*Acknowledgements.* This work has been supported by the National Project of Interest NextData by the Italian Ministry for Education, University and Research (MIUR) and by ACTRIS-2 H2020 project (G.A. 739530). Davide Putero and Luca Naitza grants have been supported by NextData. The "OpenAir" analysis package for R was obtained from http://www.openair-project.org.



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



**Table 1.** List of Essential Climate Variables (ECVs) collected at the atmospheric observatories part of NextData. Bold characters indicate the ECVs for which the automatic processing chain is already active. [1]AOD measurements are processed in the framework of AERONET and GAW-NRT programs.

| ECV | Reference programs | Monitoring stations | | | | | | |
|---|---|---|---|---|---|---|---|---|
| | | PRS | MRG | CMN | CMP | CUR | CGR | LMP |
| $CO_2$ | | X | | X | | X | X | |
| $CH_4$ | | X | | X | | X | X | |
| CO | WMO/GAW | | | X | | X | X | |
| $O_3$ | | X | **X** | **X** | X | X | **X** | **X** |
| $SO_2$ | | | | **X** | | X | X | |
| NO | | | | **X** | | X | **X** | |
| $NO_2$ | | | | **X** | | X | **X** | |
| Particle scattering | | | | **X** | | X | **X** | |
| Particle absorption | WMO/GAW, ACTRIS | | | **X** | | X | **X** | |
| Particle size distribution (by SMPS) | | | | X | | X | | |
| Particle number concentration | | | | **X** | | X | **X** | |
| Coarse particle size distribution (by OPC) | | | | **X** | X | X | **X** | |
| Aerosol Optical Depth (AOD)[1] | | | | X | | | | X |
| Meteorological parameters | WMO/GAW | X | **X** | **X** | X | X | **X** | X |
| Solar radiation | | | **X** | **X** | X | X | **X** | X |

**Table 2.** Summary of automated procedures developed for the processing of ECVs as a function of specific instruments and data file format for submission to WMO/GAW WDCs. For explanation of "data file formats" see Sect. 2.4. *Automatic processing not operative yet.

| Class | ECV | Instruments | Data file format |
|---|---|---|---|
| Trace gases (near-surface) | $O_3$ | Thermo 49i, Thermo49c | NASA-AMES |
| | NO, $NO_2$ | Thermo 42i-TL | NASA-AMES |
| | $SO_2$ | Thermo 42i-TL | NASA-AMES |
| | *CO | Picarro G2401 | ASCII (WDCGG) |
| | *$CO_2$, $CH_4$ | Picarro G2401 | ASCII (WDCGG) |
| Aerosol (near-surface) | Absorption coefficient | MAAP 5012 | NASA-AMES |
| | Scattering coefficient | TSI 3563 | NASA-AMES |
| | Particle number concentration | TSI 3775 | NASA-AMES |
| | Size distribution | Grimm 1.108 | NASA-AMES |
| Meteorology | T, RH, P, WD, WS, Global and UV radiation | Various | NASA-AMES |



**Table 3.** List of nomenclature and codes used for automated QA/QC.

| ECV class | ECV | Code |
|---|---|---|
| Trace gases | Ozone | OZO |
| | Carbon monoxide | CO |
| | Sulphur dioxide | SO2 |
| | Nitrogen oxides | NO |
| Aerosol | Scattering coefficient | NEPH |
| | Absorption coefficient | MAAP |
| | Size distribution (by OPC) | OPC |
| | Total particle number concentration | CPC |
| Meteorology | T, RH, P, WD, WS | METEO |
| | Solar radiation | RAD_SOL |

**Table 4.** Example of descriptive flags (following the NASA-AMES format) for Thermo 42i-TL instrument.

| Flag | Description |
|---|---|
| 0.000 | Valid measurements |
| 0.147 | Under detection limit (but considered valid) |
| 0.390 | Less than 50% of data used for data averaging (used for Level-2 data) |
| 0.440 | Derived value (corrected for night-time offset, valid) |
| 0.456 | Invalidated by data originator (invalid) |
| 0.664 | Low/high sampling flow, chamber pressure, PMT cooling (invalid) |
| 0.682 | Calibration or zero/span check (invalid) |
| 0.999 | Missing data (invalid) |





**Table 5.** List of parameters used for data control of Thermo 42-iTL instrument at CMN, with defined plausible threshold values. Threshold values for plausibility and variability checks vary as a function of the measurement site.

| Parameter | Variability range | Application |
| --- | --- | --- |
| Flow sample | 0.5–1.5 l/min | Diagnostic/instrumental checks |
| Chamber pressure | 200–450 mmHg | Diagnostic/instrumental checks |
| PMT cooler | $-40$–$10\,^\circ$C | Diagnostic/instrumental checks |
| Zero offset | $-0.5$–$0.5$ ppb | Diagnostic/instrumental checks |
| Span coeff. | 0.90–1.10 | Diagnostic/instrumental checks |
| Sc | 0.1–1.0 | Diagnostic/instrumental checks |
| NO variability | $NO(i)-NO(i+1) < 0.5$ ppb | Variability checks |
| NO2 variability | $NO_2(i)-NO_2(i+1) < 0.5$ ppb | Variability checks |
| NO | $> 20$ ppb | Plausibility checks |
| $NO_2$ | $> 20$ ppb | Plausibility checks |

**Table 6.** Description of data products.

| Class | Description |
| --- | --- |
| GRAPH | Time series |
| TIMEVARIATION | Average diurnal/weekly/seasonal cycles |
| CALENDAR | Daily ECV average values laid out in a calendar format |





**Figure 1.** Geographical location and pictures of the monitoring stations considered in this work.



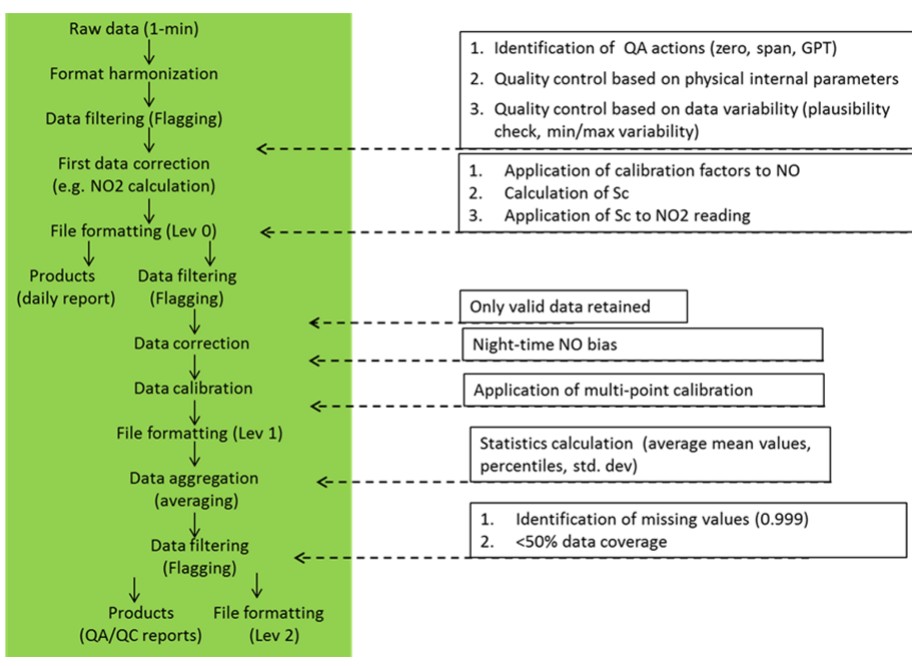

**Figure 2.** Workflow of automatic data processing for Thermo 42i-TL.



**Figure 3.** Time series of calibration factors (bkg_NO, coeff_NO, bkg_NOx, coeff_NOx, Sc) and NO-NO$_2$-NO$_x$ values during the different steps of calibration events, from January to April 2018 at CMN (NO_elab1_span, NO_elab1_gpt, NO$_2$_elab1_span, NO$_2$_elab1_gpt, NO$_2$_elab2_span, NO$_2$_elab2_zero, NO$_x$_elab2_span, NO$_x$_elab2_gpt).



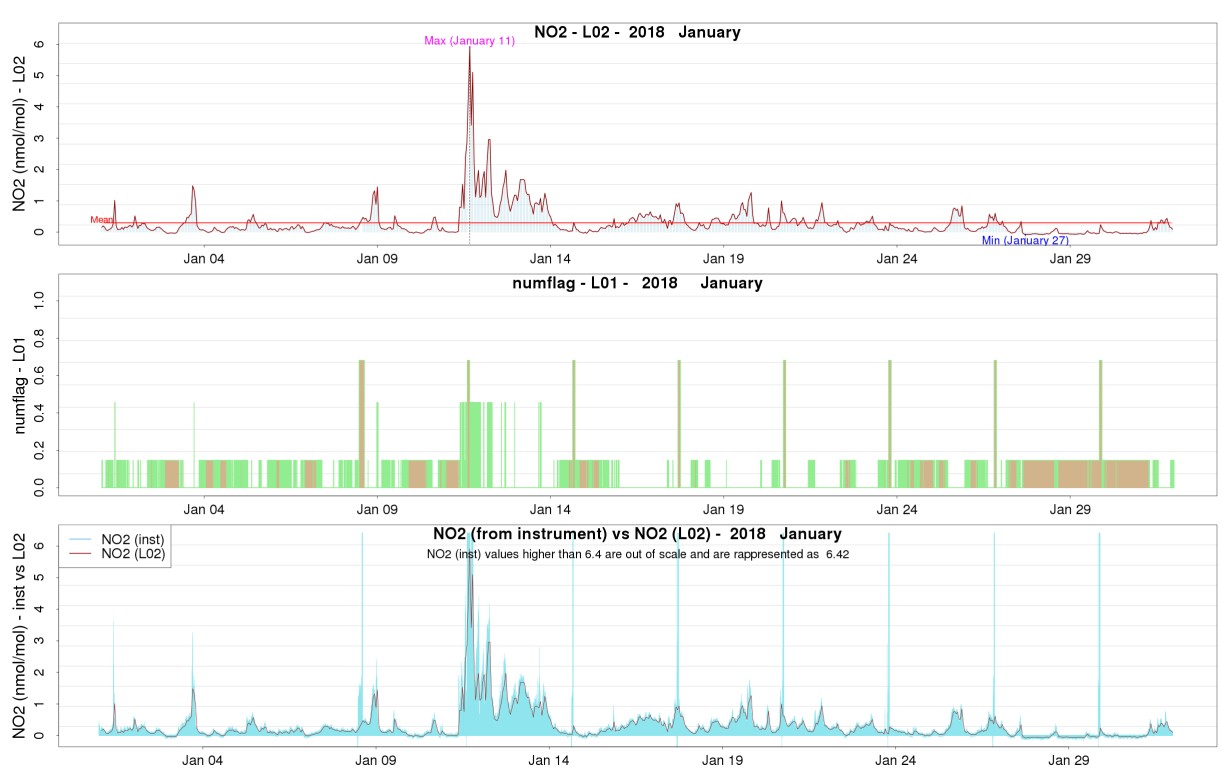

**Figure 4.** Upper: NO$_2$ (Level-2) at CMN on January 2018. The average value (red flat line) is also reported, together with the minimum (Min) and the maximum (Max) values for each month. Middle: numflag for NO$_2$ 1-min data (Level-1). Bottom: comparison between Level-2 data (red line) and raw 1-minute readings from the instrument (blue area).