# Peer review of "Automatic procedures for submitting essential climate variables (ECVs) recorded at Italian Atmospheric Observatories to WMO/GAW data centers"

_Atmospheric Measurement Techniques, 2018_

## Referee Comment (RC1) · Anonymous Referee #1 · 26 Oct 2018

The manuscript by Naitza et al. describes an automated data transmission and quality control system for a set of measurements from a set of Italian atmospheric composition monitoring stations. While this is clearly a relevant topic and I applaud the team members for having this accomplished and implemented, I don't see the manuscript worthy of publication in a peer-reviewed journal as it looks much more like a technical report. The topics of automated data processing, automated QA and flagging have been dealt with for many years, and many environmental agencies have produced heavy manuals with detailed discussion of procedures. Also WMO, which is meantioned several times in the manuscript, has produced a lot of material on such matters and operational weather centers are relying on automated procedures for daily

weather forecasts. There are practically no references to any of these long-standing activities and it is rather unclear what the novel aspects of the approach descriobed in this manuscript are.

Although I know that automated and formalized workflows are still a rare commodity in atmospheric composition science (in contrast to operational air quality monitoring), and the authors of this article are therefore among the first to develop and implement such procedures consistently throughout a set of essential climate variables, this doesn't justify the lack of any deeper analysis or discussion of the procedures. To give just one example: if range checks are performed on different atmospheric variables with different frequency distributions: how are the thresholds determined and how robust are the error detection procedures in each case? Clearly, finding outliers in, for example, NO data is very different from finding outliers in ozone, $CO_2$, or $CH_4$ data.

Also on the technical and data management side the paper lacks much important information, for example related to the documentation of responsibilities, resilience of the data transfer, provenance tracking and versioning.

In its present form this paper should only be published as technical report at one of the institution's web site. It would have to be completely rewritten to merit publication in a scientific journal - even if this journal has a more technical scope.
* * *

---

## Author Comment (AC1) · 13 Dec 2018

Here we provide a preliminary answer to the point raised by the reviewer #1. In the case of revision, we will consequently implement the manuscript. In the following, we provide our comments (AC) to referee's points.
* * *
Referee: The manuscript by Naitza et al. describes an automated data transmission and quality control system for a set of measurements from a set of Italian atmospheric composition monitoring stations. While this is clearly a relevant topic and I applaud

the team members for having this accomplished and implemented, I don't see the manuscript worthy of publication in a peer-reviewed journal as it looks much more like a technical report.

AC: We thank the reviewer for taking his/her time in reviewing our paper. We would comment that from our point of view the manuscript meet the aims and scope of this peer-reviewed journal, considering that the development and validation of techniques concerning data processing and information retrieval from gases and aerosols measurements, constitutes an important pillar for scientific activities carried out at atmospheric observatories. ———————————————————————

————————————

Referee: The topics of automated data processing, automated QA and flagging have been dealt with for many years, and many environmental agencies have produced heavy manuals with detailed discussion of procedures. Also WMO, which is mentioned several times in the manuscript, has produced a lot of material on such matters and operational weather centers are relying on automated procedures for daily weather forecasts. There are practically no references to any of these long-standing activities and it is rather unclear what the novel aspects of the approach described in this manuscript are.

AC: L. Naitza and co-authors are aware that in the framework of WMO many initiatives were carried out to deal with the question of near-real time data and automatic processes for quality check (among these WIGOS and WIS). However, the authors are pretty sure that the topic of automatic data handling and creation is still a challenge in the field of atmospheric composition data, especially for what concerns near-surface observations at atmospheric observatories. We participated to many national and international workshops, meetings and congresses: data quality check, data harmonization to specific formats and timely submission to reference data-sets are still a matter of strong concern for PIs, station managers and managing Institutions. In many situations, these processes still strongly rely on the manual intervention of technicians

and scientists. We agree that the techniques we applied for setting-up this system are probably not innovative, but a prototype system has been created and it is currently working for a subset of Italian stations. A huge technical and scientific effort has been implemented in the 2-year activity of the project to make this system working. To our knowledge, this is one of the first systematic attempts of setting-up a series of automatic procedures for some of the most diffused instruments for trace gases, aerosol properties and meteorological parameters. It is great to know that heavy manuals with detailed discussion of procedures have been produced by WMO, but at this stage these information appeared to be not diffused (or shared) in the framework of atmospheric composition science. Leading initiatives like ICOS and ACTRIS are dealing with these issues, but a number of atmospheric observatories are not part of these initiatives. For these reasons, in the framework of the National Project NextDATA, founded by the Italian Ministry of Education, Universities and Research (MIUR), to provide a contribution to a more effective participation of Italian atmospheric observatories to WMO/GAW, we proposed the system object of this paper. The authors are confident that this initiative can be useful for many other research groups in the world working on the same research field: as a matter of fact the routines developed in this work, which are the pillars of our system, will be make available with open-access policy. On the other side, we think that it is valuable to provide a detailed description of the data-creation processes which will be adopted for a subset of ECVs for specific Italian WMO/GAW observatories. In any case, we will be glad if the reviewer would provide effective contacts to systematically implement these mentioned WMO procedures to our data-sets.

———————————————————————————————————————————————

Referee: Although I know that automated and formalized workflows are still a rare commodity in atmospheric composition science (in contrast to operational air quality monitoring), and the authors of this article are therefore among the first to develop and implement such procedures consistently throughout a set of essential climate variables, this doesn't justify the lack of any deeper analysis or discussion of the procedures.

To give just one example: if range checks are performed on different atmospheric variables with different frequency distributions: how are the thresholds determined and how robust are the error detection procedures in each case? Clearly, finding outliers in, for example, NO data is very different from finding outliers in ozone, CO2, or CH4 data.

AC: Probably we were not effective enough in describing this very important topic. In the case of a paper revision, we will be able to provide the full list of automatic checks for each ECVs and single sites, both based on the internal diagnostic of instruments and on the variability of measured components (i.e., spikes or outliers detection). For the latter issue, several specific tests were carried out to define the most appropriate threshold values (below, we report an example for CPC observations at CMN, the same was done for other aerosol parameters as well as NOx). An example is shown by Figure 1. However, as stated in the manuscript, our procedures do not have the ambition to be "universal" (which is rather unfeasible considering the different span of atmospheric conditions that can affect measurements even in the same region), but they can be rather easily adapted to specific conditions and measurement sites. Thus, the thresholds used in the current prototype version of the system are specific for the ECVs and sites considered in our applications. Once the R procedures will be made available for the community, each single PI will modify them and adapt them to his need. ——————————————————————————————————————————
———————-

Referee: Also on the technical and data management side the paper lacks much important information, for example related to the documentation of responsibilities, resilience of the data transfer, provenance tracking and versioning.

AC: We think that all of these points are not within the scope of the paper. As mentioned in the manuscript, our system is a prototype (even if already running for a subset of ECVs in 4 observatories). For this reason, we did not provide all the information requested by the reviewer but, in the submitted version of the manuscript, we focus on the

description of elaboration routines. In the case of paper revision, as requested by the referee, all these information will be provided and discussed for the running prototype version of the system. ⸻

⸻

Referee: In its present form this paper should only be published as technical report at one of the institution's web site. It would have to be completely rewritten to merit publication in a scientific journal - even if this journal has a more technical scope.

AC: We are really surprised about the reviewer position. As stated in the AMT "Aims and scope", for AMT "the main subject areas comprise the development, intercomparison, and validation of measurement instruments and techniques of data processing and information retrieval for gases, aerosols, and clouds". Therefore, we think that this paper well fit with the aim of the ATM and we are confident that the explanations supplied in this letter together with the revision of the manuscript with some of the key points raised by reviewer will meet the interest of the scientific community.

⸻

**Reply to the Interactive comment on "Automatic procedures for submitting essential climate variables (ECVs) recorded at Italian Atmospheric Observatories to WMO/GAW data centers" by Luca Naitza et al. - Anonymous Referee #1 ()**

[Figure]

*An automatic data selection is applied to CPC raw data with a valid numflag (no calibration, no sampling failures/interruption). Following an approach similar to Huang et al. (2016, JORS). A sensitivity study was implemented to define the best agreement between manual outlier selection by skilled person and script results.*

**Fig. 1.**

---

## Author Comment (AC2) · 8 Jan 2019

Also on behalf of co-authors, we are glad to inform that a set of standard routines for EQC of meteorological parameters, trace gases and aerosol properties are now available at the NextDATA "Geonetwork" data repository under the "Software" resources.

http://geonetwork.igg.cnr.it/geonetwork/srv/eng/catalog.search#/search?facet.q=type%2Fsof

---

## Referee Comment (RC2) · Anonymous Referee #2 · 16 Jan 2019

This manuscript describes a suite routines to help data originators efficiently produce atmospheric variable data sets for submission to WMO/GAW Data Centers. The procedures are clearly described and well illustrated by the example of NO & NO2 data processing. It would be worth specifying if/how such a protocol facilitates the real time submission of data to the World Data Centers, since the demand for real time data is strongly increasing. Evidence that the "system" is working at the National level is also missing. This manuscript will be only marginally interesting to the scientific community until the routines described are not made available. The authors are warmly advised to ensure that this is done before the final version of the article is published.

[Figure]

Specific comments:

p. 3, line 3: Horizon2020 is EU jargon which is not understandable by non-EU readers.

p. 4, line 7: evidence that the processing chain for reactive gas and aerosol data is operational should be provided.

p. 5, line 32: "when the molecules to be quantified are scrubbed from the gas mixture" is probably what the authors mean.

p. 7, line 25: it might be useful to specify that the titration rate ("about 80%") does not affect the determination of Sc.

p. 8, line 24: why aren't data completeness < 90, 75 and 66% also flagged accordingly?

p. 9, line 8: what happens if more than 3 different numflags are encountered within a single hour? Are there prioritization rules?

p. 12, line 14: it might be specified "atmospheric background observatory network".

p. 12, line 19: even real time submission requires properly data formatting.

p. 13, line 2: station operator judgment can still be needed to flag e.g. special events like local contamination, which is not always easy to define.

---

## Author Comment (AC3) · 8 Feb 2019

Specific comments:
p. 3, line 3: Horizon2020 is EU jargon which is not understandable by non-EU readers.
*We changed with "funded by European Commission."*

p. 4, line 7: evidence that the processing chain for reactive gas and aerosol data is operational should be provided.
*A web site devoted to inspection of operational products is now available at the web address www.nextdata.isac.bo.cnr.it. In the revised manuscript, we added a paragraph (Section 3) describing this web site.*

p. 5, line 32: "when the molecules to be quantified are scrubbed from the gas mixture" is probably what the authors mean.
*Exactly, we corrected accordingly.*

p. 7, line 25: it might be useful to specify that the titration rate ("about 80%") does not affect the determination of Sc.
*OK*

p. 8, line 24: why aren't data completeness < 90, 75 and 66% also flagged accordingly?
*At the current stage, we only decided to flag data with completeness below 50%. Since numflags are provided also for data completeness below 66, 75 and 90%, this implementation can certainly be an added value in the future updates of the system.*

p. 9, line 8: what happens if more than 3 different numflags are encountered within a single hour? Are there prioritization rules?
*The one reported in the text is just an example: if more than 3 different valid numflags are encountered within the same hour, all of them must be present to form the hourly numflag. The total number of flags that can be reported is one option that can be changed by the user in the three functions that we developed. There are no prioritization rules, but, at the current stage, the reported numflags are sorted in ascending order (e.g., 0.147410640).*

p. 12, line 14: it might be specified "atmospheric background observatory network".
*Thanks for the suggestion. We implemented it, in the revised manuscript.*

p. 12, line 19: even real time submission requires properly data formatting.
*A sentence has been added in this regard: "Moreover, the products (both formatted data files and standardized plots) can be used to facilitate the activation of near-real time data delivery, as already implemented for CMN."*

p. 13, line 2: station operator judgment can still be needed to flag e.g. special events like local contamination, which is not always easy to define.
*Yes, we agree with the Reviewer on this point. Local contamination events can be selected (and properly flagged), e.g., by analyzing all of the events reported in a station logbook, or by performing a cross-analysis involving several simultaneous measurements. However, at the current stage of the system these automatic checks/comparisons are not implemented, and could certainly represent a possible future improvement for the routines. A specific comment was added on Section 4: "Our effort is expected to improve data quality, to accelerate the process of data submission to WMO/GAW WDCs (in this perspective, we will encourage the adoption of these*

*procedures also by other measurement stations, not directly related to NextData) and to make the data flagging more "objective", which means that it is based on a set of well-defined selection criteria and not uniquely related to the subjective judgment of station operators. Indeed, station operator judgment can still be needed to flag data in case of special events such as, e.g., local contamination, which is not always easy to identify by automatic tools (as showed in Sect. 2.3.1)."*

---

## Author Comment (AC4)

*We are now able to provide full answers to the issues raised by the reviewer #1. The manuscript has been implemented following the referee's comments and concerns. Once again, we would like to thank the referee for his/her revision, which was useful to better focus and describe many points of our work. In the following, we integrated (bold characters) our comments posted on 13th December 2018 (reported here in italic), for each one of the referee's points.*

Referee: The manuscript by Naitza et al. describes an automated data transmission and quality control system for a set of measurements from a set of Italian atmospheric composition monitoring stations. While this is clearly a relevant topic and I applaud the team members for having this accomplished and implemented, I don't see the manuscript worthy of publication in a peer-reviewed journal as it looks much more like a technical report.

*For answer to this general point, please refer to the author's comment published on 13th December 2018: "We thank the reviewer for taking his/her time in reviewing our paper. We would comment that from our point of view the manuscript meet the aims and scope of this peer-reviewed journal, considering that the development and validation of techniques concerning data processing and information retrieval from gases and aerosols measurements, constitutes an important pillar for scientific activities carried out at atmospheric observatories."*

Referee: The topics of automated data processing, automated QA and flagging have been dealt with for many years, and many environmental agencies have produced heavy manuals with detailed discussion of procedures. Also WMO, which is mentioned several times in the manuscript, has produced a lot of material on such matters and operational weather centers are relying on automated procedures for daily weather forecasts. There are practically no references to any of these long-standing activities and it is rather unclear what the novel aspects of the approach described in this manuscript are.

*Author's comment by 13th December 2018: "L. Naitza and co-authors are aware that in the framework of WMO many initiatives were carried out to deal with the question of*

*near-real time data and automatic processes for quality check (among these WIGOS and WIS). However, the authors are pretty sure that the topic of automatic data handling and creation is still a challenge in the field of atmospheric composition data, especially for what concerns near-surface observations at atmospheric observatories. We participated to many national and international workshops, meetings and congresses: data quality check, data harmonization to specific formats and timely submission to reference data-sets are still a matter of strong concern for PIs, station managers and managing Institutions. In many situations, these processes still strongly rely on the manual intervention of technicians and scientists. We agree that the techniques we applied for setting-up this system are probably not innovative, but a prototype system has been created and it is currently working for a subset of Italian stations. A huge technical and scientific effort has been implemented in the 2-year activity of the project to make this system working. To our knowledge, this is one of the first systematic attempts of setting-up a series of automatic procedures for some of the most diffused instruments for trace gases, aerosol properties and meteorological parameters. It is great to know that heavy manuals with detailed discussion of procedures have been produced by WMO, but at this stage these information appeared to be not diffused (or shared) in the framework of atmospheric composition science. Leading initiatives like ICOS and ACTRIS are dealing with these issues, but a number of atmospheric observatories are not part of these initiatives. For these reasons, in the framework of the National Project NextDATA, founded by the Italian Ministry of Education, Universities and Research (MIUR), to provide a contribution to a more effective participation of Italian atmospheric observatories to WMO/GAW, we proposed the system object of this paper. The authors are confident that this initiative can be useful for many other research groups in the world working on the same research field: as a matter of fact the routines developed in this work, which are the pillars of our system, will be make available with open-access policy. On the other side, we think that it is valuable to provide a detailed description of the data-creation processes which will be adopted for a subset of ECVs for specific Italian WMO/GAW observatories. In any case, we will be glad if the reviewer would provide effective contact able to systematically implement these mentioned WMO procedures to our data-sets."*

**Comment update: "The authors would like to stress that references to ICOS-RI, which implemented an operative system for EQC of a selected number of parameters related with the carbon cycle investigation (i.e. CO, $CO_2$, $CH_4$, $N_2O$ and meteorological parameters), were already present in the paper. We further stress these references in**

*the revised version of the paper. Moreover, references to the automated QA/QC system implemented in the framework of GMOS for mercury measurements were already available in the original version of the manuscript.*

*In the revision of the manuscript, the following sentence was reported in the "Introduction" section: "Examples of systems for the automatic execution of QA/QC activities in the framework of atmospheric composition ECVs can be found within ICOS and GMOS initiatives. As described by Hazan et al., (2016), the former is related to the Evaluation and Quality Control (EQC) of a relevant subset of atmospheric parameters related with the carbon cycle investigation (i.e. CO, $CO_2$, $CH_4$, $N_2O$ and meteorological parameters), while the latter implemented a web-based system for QA/QC to check raw data related to atmospheric mercury (D'Amore et al. 2015)".*

Although I know that automated and formalized workflows are still a rare commodity in atmospheric composition science (in contrast to operational air quality monitoring), and the authors of this article are therefore among the first to develop and implement such procedures consistently throughout a set of essential climate variables, this doesn't justify the lack of any deeper analysis or discussion of the procedures. To give just one example: if range checks are performed on different atmospheric variables with different frequency distributions: how are the thresholds determined and how robust are the error detection procedures in each case? Clearly, finding outliers in, for example, NO data is very different from finding outliers in ozone, CO2, or CH4 data.

*Author's comment by 13[th] December 2018: "Probably we were not effective enough in describing this very important topic. In the case of a paper revision, we will be able to provide the full list of automatic checks for each ECVs and single sites, both based on the internal diagnostic of instruments and on the variability of measured components (i.e., spikes or outliers detection). For the latter issue, several specific tests were carried out to define the most appropriate threshold values (below, we report an example for CPC observations at CMN, the same was done for other aerosol parameters as well as NOx).*

[Figure]

*An automatic data selection is applied to CPC raw data with a valid numflag (no calibration, no sampling failures/interruption). Following an approach similar to Huang et al. (2016, JORS). A sensitivity study was implemented to define the best agreement between manual outlier selection by skilled person and script results.*

*However, as stated in the manuscript, our procedures do not have the ambition to be "universal" (which is rather unfeasible considering the different span of atmospheric conditions that can affect measurements even in the same region), but they can be rather easily adapted to specific conditions and measurement sites. Thus, the thresholds used in the current prototype version of the system are specific for the ECVs and sites considered in our applications. Once the R procedures will be made available for the community, each single PI will modify them and adapt them to his need."*

***Comment update: "We now provided in the supplementary material the complete list of threshold values and methods currently implemented in the prototype system for the data automatic flagging. Moreover, a specific section (Section 2.3.1) has been added for dealing with the topic of the automatic outlier selection. Many comments have been added about this topic. Finally, in agreement with our commented posted on January 2019, the following sentences have been added to the revised manuscript (Sect. 4): "Concerning the definition of the threshold values to be adopted for the automatic flagging, it should be clearly stated that the current set of values does not have the ambition to be "universal" (which is rather unfeasible considering the different span of atmospheric conditions that can affect measurements even in the same region), but they are specific for the ECVs and sites considered in this prototype application. However, they can be rather easily changed and adapted to specific conditions and measurement sites: each single PI will modify them and adapt them to his/her specific needs. However, a not negligible amount of work has been carried out also in the direction of setting-up automatic functions for outlier selection (see Sect. 2.3.1), which are already implemented in the processing chain of aerosol parameters.***

*A sensitivity test was carried out to assess the impact of adopting different outlier selection methods and settings to the time series of a subset of ECVs (NO, eqBC, and total particle number concentration) and to the representation of their averaged seasonal diurnal variabilities."*

Also on the technical and data management side the paper lacks much important information, for example related to the documentation of responsibilities, resilience of the data transfer, provenance tracking and versioning.

*Author's comment by 13th December 2018: "We think that all of these points are not within the scope of the paper. As mentioned in the manuscript, our system is a prototype (even if already running for a subset of ECVs in 4 observatories). For this reason, we did not provide all the information requested by the reviewer but, in the submitted version of the manuscript, we focus on the description of elaboration routines. In the case of paper revision, as requested by the referee, all these information will be provided and discussed for the running prototype version of the system."*

**Comment update: "In the revised version of the manuscript we provided the requested information. A new section (Sect. 3) has been added to describe the web interface for visualization of created data product. In particular, in the revised Section 2.1 we reported: "All the developed routines are virtually stand-alone and any hypothetical user, after installing "R", which is a free software environment for computing and graphics, can use them on his/her own PC or server, for both automatic and on-demand applications. Thus, our software is characterized by a good level of portability, useful, as an instance, for migrating or installing to different computer systems. To favor software usability, portability and understanding by operators, each script is accompanied with a header where basic instructions for installation, usage and modification, along with code update history, are provided.**
**With the purpose of demonstrating the effectiveness of the proposed routines and supporting Italian observatories in the process of data production and submission, a centralized prototype system was implemented for the atmospheric observatories 5 involved within the NextData project.**
**While, as detailed below, the delivery of data files from the atmospheric observatories is a duty in charge of observatory personnel/Institutions, the automatic operation of the routines is in charge of a small group of people at the CNR–ISAC HQs in Bologna, involving an IT expert and two routine developers. On a short to medium time scale this would assure the scientific and expert operational actions, as well as**

*the programmatic support to underpin the system. In case the raw data formats are not changed, the system is expected to be robust and frequent technical intervention is not needed. The interrupted or delayed data flow from the observatories does not represent an issue, since the routines are designed to run on each calendar day, and to process all the synchronized available data files in the current year. To facilitate the access to the data products by the observatories PIs, a web interface was activated (http://nextdata.bo.isac.cnr.it, see Sect. 3).*

*To check the correct execution of the routines, a specific product (called "health status report") was designed to be accessed 15 by measurement PIs or CNR–ISAC personnel. This product is generated on a daily basis and the plots indicate, for each observatory, the correct execution of elaboration routines (see Fig. 3 for an example): if a routine worked successfully, a bar is drawn (in case of routine failure, the related bar is not plotted).*

*As stated above, we rely on the strong assumption that the responsibility for the production of "final" consolidated data files (to be submitted to reference WMO/GAW data centers or simply to be published for external usage) is totally in charge of the measurement PIs. At the moment of data submission or publication, PIs are expected to review the result of the automatic flagging produced by the routines at the lowest data level (i.e., Level-0) and they can accept or modify the produced file. In the latter case, it is recommended that a new file version is created (the NASA-Ames format requires to declare the revision number as well as the date creation of each data file) and that, based on the revised (consolidated) Level-0 file, new versions of Level-1 and Level-2 files are created using "P21" and "P22" routines."*

In its present form this paper should only be published as technical report at one of the institution's web site. It would have to be completely rewritten to merit publication in a scientific journal - even if this journal has a more technical scope.

*Here, we would like to present here again our reply already posted on 13[th] December 2018. "We are really surprised about the reviewer position. As stated in the AMT "Aims and scope", for AMT "the main subject areas comprise the development, intercomparison, and validation of measurement instruments and techniques of data processing and information retrieval for gases, aerosols, and clouds". Therefore, we think that this paper well fit with the aim of the ATM and we are confident that the explanations supplied in this letter together with the revision of the manuscript with some of the key points raised by reviewer will meet the interest of the scientific community."*

*We would like to add that the paper has been implemented with a number of new analyses (e.g. outlier screening) and technical details about our system (sustainability and software readiness) and developed procedures (e.g. methodology for correcting baseline drift of $SO_2$). These procedures (R scripts) are now freely available from the NextData Geonetwork system and the web link for downloading the scripts is now available. The routines can be integrated and, in case, optimized by any external user.*